# Mechanism of ADP-Inhibited ATP Hydrolysis in Single Proton-Pumping F_o_F_1_-ATP Synthase Trapped in Solution

**DOI:** 10.3390/ijms24098442

**Published:** 2023-05-08

**Authors:** Iván Pérez, Thomas Heitkamp, Michael Börsch

**Affiliations:** Single-Molecule Microscopy Group, Jena University Hospital, 07743 Jena, Germany; ivan.perezguzman@med.uni-jena.de (I.P.); thomas.heitkamp@med.uni-jena.de (T.H.)

**Keywords:** F_o_F_1_-ATP synthase, ATP hydrolysis, ADP inhibition, single-molecule FRET, ABEL trap, subunit rotation, catalytic mechanism

## Abstract

F_o_F_1_-ATP synthases in mitochondria, in chloroplasts, and in most bacteria are proton-driven membrane enzymes that supply the cells with ATP made from ADP and phosphate. Different control mechanisms exist to monitor and prevent the enzymes’ reverse chemical reaction of fast wasteful ATP hydrolysis, including mechanical or redox-based blockade of catalysis and ADP inhibition. In general, product inhibition is expected to slow down the mean catalytic turnover. Biochemical assays are ensemble measurements and cannot discriminate between a mechanism affecting all enzymes equally or individually. For example, all enzymes could work more slowly at a decreasing substrate/product ratio, or an increasing number of individual enzymes could be completely blocked. Here, we examined the effect of increasing amounts of ADP on ATP hydrolysis of single *Escherichia coli* F_o_F_1_-ATP synthases in liposomes. We observed the individual catalytic turnover of the enzymes one after another by monitoring the internal subunit rotation using single-molecule Förster resonance energy transfer (smFRET). Observation times of single FRET-labeled F_o_F_1_-ATP synthases in solution were extended up to several seconds using a confocal anti-Brownian electrokinetic trap (ABEL trap). By counting active versus inhibited enzymes, we revealed that ADP inhibition did not decrease the catalytic turnover of all F_o_F_1_-ATP synthases equally. Instead, increasing ADP in the ADP/ATP mixture reduced the number of remaining active enzymes that operated at similar catalytic rates for varying substrate/product ratios.

## 1. Introduction

F_o_F_1_-ATP synthases catalyze ATP synthesis from ADP and phosphate. These universal membrane-embedded enzymes are found in bacteria, chloroplasts, and mitochondria [1,2]. The energy stored in the electrochemical potential of protons (or other ions in some organisms) across the membrane comprises two components, i.e., a proton concentration difference, ΔpH, and an electric potential, Δψ, and is called the proton motive force (PMF) [3]. ATP synthesis rates depend on the PMF as the driving force as well as the local concentrations of ADP, phosphate, and ATP.

The molecular mechanism of ATP synthesis involves three synchronized, sequentially operating catalytic nucleotide binding sites on the F_1_ domain (subunit composition α_3_β_3_γδε for the *E. coli* enzyme) and the internal stepwise rotation of the γ- and ε-subunits [4,5]. These two subunits are connected to and driven by the rotating ring of 10 c-subunits in the membrane-embedded F_o_ domain. To harness the PMF, proton translocation through the transmembrane a-subunit of the F_o_ domain (subunit composition ab_2_c_10_) requires entering one proton after another into the half-channel that is open towards the high-concentration side of protons. Within the a-subunit, the proton is transferred to the adjacent c-subunit harboring an empty proton-binding site, and this causes a single 36°-stepped rotation of the complete c-ring in the F_o_ domain. Thereby, another c-subunit approaches the exit half-channel for protons in the *a*-subunit, where a proton is transferred and then leaves the a-subunit to the other side of the membrane with the lower H^+^ concentration [6]. The direction of c-ring rotation is “clockwise” when viewed from the membrane towards to F_1_ domain [7,8]. Accordingly, the mechanics of the c-ring rotary motor have been compared to a proton-driven turbine.

When the PMF drops below a threshold, F_o_F_1_-ATP synthases can catalyze the opposite chemical reaction, i.e., ATP hydrolysis-driven subunit rotations in F_1_ and the coupled F_o_ domains occur in reverse and are associated with active proton pumping across the lipid membrane [9]. Because the F_1_ domain can be stripped from the F_o_ domain by biochemical methods, the buffer-soluble F_1_ domain can be investigated as an ATP-hydrolyzing enzyme in the absence of a PMF. ATP binding to the empty nucleotide binding site results in a conformational change of the corresponding β-subunit, and this closing motion pushes the adjacent part of the asymmetric γ-subunit into a rotary motion of 120°. The direction of ATP-driven γ- and ε-subunit rotation is “counter-clockwise” when viewed from the membrane side to the top of F_1_ [10].

The asymmetric shape of the central γ-subunit defines which of the three binding sites is open and empty and can bind ATP, which is closed and hydrolyzes ATP, and which is half-closed with the products ADP and phosphate to be released, respectively. The 1997 Nobel Prize in Chemistry was awarded in part to P. D. Boyer for the concept of the rotary “binding change mechanism” [3] and to J. E. Walker for the first high-resolution structure of the mitochondrial α_3_β_3_γ F_1_-subcomplex unraveling the relations of γ-subunit asymmetry, binding site conformations and nucleotide occupancies [11].

ATP-driven γ-subunit rotation was revealed by biochemical [12] and spectroscopic [13] approaches using the F_1_ domain. However, cuvette-based ensemble measurements cannot discriminate between active and inactive enzymes, and any kinetic information is always averaged over the subpopulations of active and inhibited F_1_ domains. Introducing videomicroscopy of single, surface-attached α_3_β_3_γ F_1_-subcomplexes in 1997 allowed H. Noji and coworkers to directly visualize the ATP-driven γ-subunit rotation in the “counter-clockwise” direction using a specifically attached fluorescent marker [10]. They also determined the low percentage of actively rotating versus non-rotating F_1_-subcomplexes in the microscopic field-of-view and analyzed catalytic turnover based on the active rotary motors on the surface. In the following two decades, refined single-molecule rotation assays provided a plethora of new insights into the stepsizes of the rotor, angular-resolved catalytic sub-processes, motor properties such as torque and speed, and inhibitory controls by mechanical blockade of rotation through the C-terminus of the ε-subunit and by ADP product inhibition (for reviews see [14,15,16,17]).

We have introduced a different single-molecule fluorescence approach to study the sequential conformational changes of the complete membrane-embedded F_o_F_1_-ATP synthase in liposomes one by one [18]. Two distinct fluorophores are attached specifically to the enzyme, one fluorophore to either of the three rotary subunits γ, ε or c, and a second fluorophore to one of the stator subunits (a-, β- or b-subunits). Intramolecular distance recordings between these two markers rely on Förster resonance energy transfer (smFRET) for the range of 2 to 10 nm. With smFRET we could show that the direction of γ- and ε-subunit rotation is reversed when switching from ATP hydrolysis to ATP synthesis conditions [19,20], that the stepsize of the c-ring rotation is 36° during proton-driven ATP synthesis [21], and that an elastic deformation of the γ,ε rotor occurs with respect to the c-ring in order to compensate for the symmetry mismatch between the three-stepped and ten-stepped coupled motors [22]. However, our observation times of freely diffusing single proteoliposomes through an engineered, enlarged confocal detection volume were still significantly shorter than one second. In addition, the strong fluorescence intensity fluctuations due to Brownian motion limited the precision of distance calculations in the time traces of FRET-labeled single F_o_F_1_-ATP synthases.

Therefore, we aimed to increase the observation times and minimize intensity fluctuations by using a fast confocal anti-Brownian electrokinetic trap (ABEL trap), a single-molecule detection device introduced in 2005 by A. E. Cohen and W. E. Moerner [23,24]. This confocal microscope setup uses microfluidics to confine diffusion in two dimensions. It locates a fluorescent nanoparticle in the trap region by generating a fast-switching laser focus pattern for position-correlated photon detection. By calculating and applying appropriate voltages to four Pt-electrodes in real-time, the fluorescent nanoparticle is pushed back always into the center of the laser focus pattern and it can be monitored until photobleaching occurs. The ABEL trap provides the highest-possible resolution for smFRET-based intramolecular distance measurements [25] and intensity-based fluorophore counting [26], as well as for single fluorophore diffusion analysis [27,28,29,30] and electric charge determination in solution [31].

Here, we applied our confocal ABEL trap setup [32] to hold FRET-labeled single F_o_F_1_-ATP synthases in focus and to record smFRET time traces in solution. With trap times of up to a second, we could clearly discriminate and count individual actively rotating enzymes and non-rotating inhibited ones. Varying the ADP/ATP ratio from 0/100 μM to 100/0 μM allowed us to measure both the fraction and turnover of active F_o_F_1_-ATP synthases. We found similar catalytic rates of the active enzymes for almost all ADP/ATP ratios, at least for ATP contents of 30 μM and higher. However, the percentage of active enzymes decreased with respect to the relative ADP content in the nucleotide mixture. In contrast to the averaging approaches of biochemical assays, our single-molecule enzymology approach provided the mechanistic insights into product inhibition affecting individual enzymes differently over time.

## 2. Results

### 2.1. Monitoring Catalysis-Related ε-Subunit Rotation by Single-Molecule FRET

To count active F_o_F_1_-ATP synthases showing ATP-driven subunit rotation and to discriminate from inhibited inactive enzymes, a single-molecule FRET assay was established. First, we labeled the rotary ε-subunit at residue position 56 with the fluorophore Cy3B. Therefore, we previously introduced a cysteine mutant (εH56C) and purified the F_1_ domain from the *E. coli* enzyme [20]. The maleimide derivative of Cy3B was used for specific labeling of the rotor subunit. Next, a second fluorophore, Alexa Fluor 647, was attached to the a-subunit of the F_o_ domain using a mutant with an extended C-terminus comprising a cysteine at a residue position 276 [33]. We purified the F_o_F_1_-ATP synthase mutant from *E. coli*, labeled the a-subunit, reconstituted the labeled F_o_F_1_-ATP synthase in liposomes, and exchanged the F_1_ domain with the Cy3B-labeled F_1_ domain. Details are given in Section 4 below and in reference [32].

Alexa Fluor 647 on the static *a*-subunit served as the acceptor fluorophore for intramolecular distance measurements by Förster resonance energy transfer to the donor Cy3B. The Förster radius for the FRET pair of fluorophores is R_0_ = 7.2 nm for a 50% energy transfer (see https://www.fpbase.org/FRET; accessed on 18 April 2023). Laser excitation of Cy3B was achieved with 532 nm in continuous-wave mode in our confocal microscope. Briefly, the feedback-controlled microfluidics of our custom-designed confocal anti-Brownian electrokinetic trap (ABEL trap) microscope prolonged the observation times of single FRET-labeled F_o_F_1_-ATP synthase by counter-acting the diffusive motion of the trapped proteoliposome in real-time. The ABEL trap allowed us to minimize fluorescence fluctuations by achieving spatiotemporal homogeneous excitation of and detection from the 2.34 × 2.34 μm^2^ trapping region. Therefore, the FRET-based intramolecular distance measurements between Cy3B and Alexa Fluor were recorded for one enzyme after another, with time traces reaching more than one second until photobleaching of the fluorophores.

For unequivocal detection of subunit rotation in F_o_F_1_-ATP synthase, the intramolecular marker distances had to change significantly, i.e., in the nanometer range. Very recent cryoEM structures of the three major rotary conformations of the *E. coli* enzyme in the presence of ATP allowed us to estimate the expected marker positions (Figure 1) [34,35,36].

The expected Cy3B-Alexa Fluor 647 distances for the inactive, ADP-inhibited enzyme were modeled according to the cryoEM structures of the detergent-solubilized F_o_F_1_-ATP synthase in the presence of 10 mM ADP (Figure 1A–C) [35]. The cryoEM structures revealed three major rotary states with the C-terminus of ε-subunit in the rotation-blocking “up” configuration. However, each of the states #1, #2, and #3 showed some distinguishable conformational substates. Especially the substates of state #1 differed significantly in the relative position of their a-subunit, i.e., the likely position of our C-terminal extension. Here, we decided to use substates #1C (PCSB protein data bank entry PDB 6oqt), #2 (PDB 6oqv), and #3 (PDB 6oqw) to model the distances between the cysteines ε56C and a276C. Accordingly, the rotation-blocked state #1 showed a distance of 3.2 nm, state #2 showed 7.6 nm and state #3 showed 5.0 nm. Given the Förster radius and an additional linker length, we anticipated finding only two distinct FRET efficiencies for the ADP-inhibited F_o_F_1_-ATP synthases. We expected either one high FRET state (corresponding to one of the two similar high FRET states in the ε-CTD “up” conformations, Figure 1A–C) or one medium FRET state (Figure 1B). For the rotation-inhibited enzymes, these FRET states would remain constant during the observation time of each ABEL-trapped proteoliposome.

Next, we built three-dimensional models for the active ATP-hydrolyzing enzyme with the regulatory C-terminus of the ε-subunit (ε-CTD) in the “down” configuration (Figure 1D–F) [36] and calculated the three distances between the cysteines ε56C and a276C. For the modeled rotary state #1 (PDB 8dbp), the distance was 1.7 nm; for state #2 (PDB 8dbt), it was 6.3 nm; and for state #3 (PDB 8dbv), it was 7.0 nm. Because the fluorophores are bound to the S atoms of the cysteines by a chemical linker chain, additional lengths of 0.5 to 1 nm should be added to these three distances. Given the Förster radius R_0_ of 7.2 nm, we anticipated to find only two distinguishable FRET efficiencies in the time traces of active F_o_F_1_-ATP synthases, i.e., a medium FRET efficiency around 0.5 representing the two long distances for the rotary states #2 and #3, and a very high FRET efficiency for state #1.

### 2.2. Constant and Fluctuating FRET States at Different ADP/ATP Ratios

For the discrimination of fluctuating from constant FRET states in a photon burst, we used the unprocessed values of the proximity factor P instead of the instrument- and photophysics-corrected FRET efficiency E_FRET_. P was calculated from the photon counts in the donor and acceptor channel per data point as P = I_A_/(I_D_ + I_A_), with I_A_ being photon counts of the acceptor channel and I_D_ being photon counts of donor channel, within 1 ms binning. The varying background was subtracted in both donor and acceptor channels as detected in the vicinity of the selected photon burst.

Figure 2 shows examples of F_o_F_1_-ATP synthases in the presence of 30 μM ADP and 70 μM ATP. Both FRET-labeled active enzymes exhibited very distinct ATP turnover. The F_o_F_1_-ATP synthase in Figure 2A was trapped at a laboratory time of about 330 ms as characterized by a sudden increase of the photon counts in the donor (green) and the acceptor (magenta) channel. This enzyme was trapped for 495 ms. The sum of both donor and acceptor photon counts is shown as the gray intensity trace. The proximity factor P trace (light blue, upper panel) fluctuated between two FRET states according to the Hidden-Markov Model (HMM) assignment by the DISC software [37]. For details, see Section 4 below. The number of fluctuations of the proximity factor was analyzed by assuming that a full 360° rotation of the ε-subunit was characterized by a pair of two distinct FRET states (i.e., one high plus one low FRET state, or vice versa). Each FRET state pair corresponded to three hydrolyzed ATP molecules. Thus, the enzyme in Figure 2A with 26 FRET-state pairs hydrolyzed 78 ATP molecules in 495 ms, i.e., with a mean ATP hydrolysis rate of 168 ATP/s.

In contrast, the F_o_F_1_-ATP synthase shown in Figure 2B hydrolyzed ATP much slower. Trapped as an active enzyme for about 810 ms, the four FRET state pairs corresponded to a mean ATP hydrolysis rate of 21 ATP/s. The phenomenon of different catalytic activities of individual enzymes is often called “*static disorder*” in single-molecule enzymology [38].

Next, we asked whether the very different ATP turnover numbers were time-dependent with respect to the start of the smFRET recordings after ADP/ATP addition. Over the total time course of 30 min recordings, we detected active enzymes with different mean catalytic turnover. No evidence was found that F_o_F_1_-ATP synthases were hydrolyzing ATP with different speeds at the beginning and at the end of the recordings (Figure 2C). The average ATP hydrolysis rate calculated from the mean of each enzyme was 88 ATP/s in the presence of 30 μM ADP and 70 μM ATP.

Alternatively, we calculated the average ATP hydrolysis rate by combining the dwell times of all FRET pairs from the active enzymes and fitting the distribution with a monoexponential decay function (Figure 2D). The fit to the histogram yielded a 1/e time of 21.5 ms. Assuming each full rotation of the ε-subunit in F_o_F_1_-ATP synthase corresponded to three hydrolyzed ATP molecules, an average turnover of 139 ATP/s was deduced for this 30/70 ADP/ATP ratio.

Similarly, ATP-driven ε-subunit rotation was analyzed in the presence of 70 μM ADP and 30 μM ATP. Shown in Figure 3A,B are the time traces of two active FRET-labeled enzymes. The F_o_F_1_-ATP synthase in Figure 3A was trapped at about 60 ms and showed fast FRET fluctuations until photobleaching of the acceptor Alexa Fluor 647 occurred at a laboratory time around 440 ms. As a result of the remaining “donor only” signal, the enzyme remained in focus hold by the ABELtrap for another 400 ms, or it was trapped for 800 ms in total, respectively. During the period of FRET fluctuations, 39 ATP molecules were hydrolyzed in 358 ms corresponding to an ATP turnover of 109 ATP/s. The second F_o_F_1_-ATP synthase in Figure 3B was trapped at about 250 ms and rotated for 154 ms, exhibiting a mean hydrolysis rate of 58 ATP/s.

This nucleotide mixture contained a much larger fraction of the catalytic product ADP. However, the smaller amount of ATP was still high enough that a significant substrate consumption over the time course of 30-min recordings was not observed, and slow and fast rotating enzymes were detected with similar probabilities at the beginning and the end of the measurements (Figure 3C). The average ATP hydrolysis rate calculated from the mean of each individual enzyme was 86 ATP/s (red line in Figure 3C).

Alternatively, the average ATP hydrolysis rate was calculated by combining the dwell times of all FRET pairs from all active enzymes. Fitting the distribution with a monoexponential decay function yielded an average turnover of 106 ATP/s (Figure 3D). The fit to the histogram yielded a 1/e time of 28.2 ms. These turnover numbers were similar to the conditions of 30 μM ADP plus 70 μM ATP above. However, the number of active enzymes and the total number of FRET state pairs were significantly lower in the nucleotide mixture with lower ATP content.

### 2.3. Similar ATP Turnover Found for All ADP/ATP Ratios

We systematically examined different nucleotide mixtures with a total nucleotide concentration of 100 μM, i.e., the ADP/ATP ratios of 0/100, 10/90, 20/80, 30/70 (see Figure 2), 40/60, 50/50, 60/40, 70/30 (see Figure 3), 80/20, 90/10 and 100/0. For all conditions we found F_o_F_1_-ATP synthases showing FRET state fluctuations as assigned by the applied HMM software. Examples of fast and of slow ε-subunit rotations at all ADP/ATP ratios are given in the Appendix A.

The mean catalytic activities of the enzymes did not correlate with the ATP or the ADP content of the nucleotide mixture as seen in Figure 4. For all conditions, the mean turnover calculated from each individual active enzyme was 88 ± 15 ATP/s. We note that the ADP/ATP ratios 80/20, 90/10, and 100/0 were excluded from an interpretation analysis due to the very low number of active enzymes in these three data sets. While the mean turnover did not vary, the number of active enzymes detected in the measurements dropped significantly from 49 enzymes for the 0/100 ADP/ATP ratio (Figure 4A) down to two enzymes for the 80/20 ADP/ATP ratio or three enzymes for the 90/10 ADP/ATP ratio (Figure 4J). In the absence of ATP, i.e., for the 100/0 ADP/ATP ratio, two enzymes with apparently fluctuating FRET efficiencies were considered by the DISC HMM software. All mean turnover numbers are summarized in Appendix A.

Alternatively, we analyzed the distributions of the lengths (or dwells) of FRET state pairs in active enzymes for all 11 ADP/ATP conditions (Figure 5). As stated above, each pair of one high FRET state plus a subsequent medium FRET state corresponded to a full rotation, or to three hydrolyzed ATP molecules, respectively. Fitting the histograms yielded similar average turnover numbers. As discussed above, the ADP/ATP ratios 80/20, 90/10, and 100/0 were excluded from this analysis due to the very low number of active enzymes in these data sets. The slowest average turnover was 99 ATP/s for the 20/80 ADP/ATP ratio (Figure 5C) and the fastest was 140 ATP/s for the 30/70 ADP/ATP ratio (Figure 5D). The average turnover was 114 ATP/s. Combining the FRET state pair durations of all active enzymes recorded at one ADP/ATP ratio and fitting these distributions resulted in slightly higher average catalytic rates than calculating the individual mean turnover per enzyme by averaging the individual rates first. As seen in Figure 5, the total number of FRET state pairs decreased from the 0/100 ADP/ATP ratio with the highest ATP concentration to the ADP/ATP ratio with the lowest ATP concentration. All average turnover numbers are summarized in Appendix A.

### 2.4. Similar Distributions of the FRET State Fluctuation Periods for All ADP/ATP Ratios

Next, we considered the possibility that the competing ADP binding might affect or shorten the time periods when the F_o_F_1_-ATP synthase is actively hydrolyzing ATP. The duration of FRET state fluctuations for each individual active enzyme was plotted with respect to the ADP/ATP ratio. The box plot is shown in Figure 6 and the scatter plot for these data is found in the Appendix A.

For an ADP/ATP ratio of 50/50 and for lower relative amounts of ADP in the nucleotide mixture, the mean durations of the fluctuating FRET state periods were almost identical in a range of 200 to 300 ms (the median as well as the arithmetic mean, see right part of Figure 6). The box plot analysis provided the upper limit for 90% of the individual fluctuating FRET state periods in the range of 400 to 500 ms. However, many of the fluctuating FRET state periods were much shorter and in the range of 100 ms (i.e., the lower 10% limit of active ATP turnover durations in the box plots). For the lowest ATP contents in the ADP/ATP ratios 100/0, 90/10, and 80/20, the number of active F_o_F_1_-ATP synthases was less than 5, and, therefore, an interpretation of these few enzymes was omitted (shaded box plots in the left part of Figure 6).

We compared the durations of the fluctuating FRET state periods with the durations of the FRET states of non-rotating inactive F_o_F_1_-ATP synthases obtained from the same smFRET data sets. As an additional reference, the durations of the fluctuating FRET state periods for active enzymes in the presence of 5 different ATP concentrations but without ADP were re-analyzed from previously recorded smFRET data [32]. The related scatter plots are shown in Appendix A (active enzymes in the presence of ATP only), Appendix A (inactive enzymes with ADP/ATP) and Appendix A (inactive enzymes with ATP only) in the Appendix A. Accordingly, durations of active ATP hydrolysis as well as durations of catalytic inactivity were very similar and were ranging from 50 to 500 ms. Because of these similarities, it could not be ruled out that the observed limited durations of FRET states were probably caused by photobleaching, most likely of the FRET acceptor Alexa Fluor 647 in the absence of an oxygen scavenger system and triplet quencher in the buffer.

### 2.5. Decreasing Percentage of Active F_o_F_1_-ATP Synthases with Higher ADP Content

Finally, we compared the activities of FRET-labeled F_o_F_1_-ATP synthases in the presence of ADP/ATP mixtures with our previously recorded smFRET data of the same enzyme preparation measured in the presence of different ATP concentrations but without ADP [32]. We re-analyzed the “ATP only” smFRET data for ATP concentrations of 5 μM, 20 μM, 40 μM, 100 μM, and 1000 μM using the DISC HMM software [37] to assign FRET states and mean individual ATP hydrolysis rates for the active enzymes.

Figure 7A,B show the side-by-side comparison as box plots. For all ADP/ATP with 30 μM ATP and higher ATP content, the average of the mean ATP turnover stayed constant (Figure 7A). The data for ADP/ATP ratios with lower ATP content were not considered due to the total number of active enzymes being less than five. When only ATP was present to drive catalysis, the average of the ATP turnover remained constant for ATP concentrations of 40 μM and higher, but dropped for lower ATP concentrations (Figure 7B). The ATP concentration dependence for catalytic turnover has been fitted previously according to Michaelis-Menten kinetics to yield a Michaelis-Menten constant K_M_ = 15 ± 3 μM ATP for the half-maximal ATP turnover of the FRET-labeled F_o_F_1_-ATP synthases in liposomes [32]. The re-analyzed smFRET data showing the mean turnover of individual FRET-labeled F_o_F_1_-ATP synthases for the five ATP concentrations are shown in the Appendix A, and the re-analyzed average turnover based on FRET state pairs of all active F_o_F_1_-ATP synthases for the five ATP concentrations are found in the Appendix A.

As the total number of FRET state pairs decreased with increasing amounts of ADP, or decreasing amounts of ATP, respectively, we counted the numbers of inactive F_o_F_1_-ATP synthases characterized by a constant medium FRET or a high FRET state for each ADP/ATP ratio and calculated the percentage of active enzymes for all conditions. As shown in Figure 7C, the maximum percentage of active enzymes was 25% at low ADP content and dropped down to 8% for the nucleotide mixture of 70 μM ADP and 30 μM ATP. In contrast, the percentage of active enzymes in the presence of ATP only did not change significantly for lower ATP concentrations (Figure 7D). About 30% of the FRET-labeled F_o_F_1_-ATP synthases were actively hydrolyzing ATP at any ATP concentration, but 70% of the enzymes were found inactive and did not exhibit FRET state fluctuations.

## 3. Discussion

It has been known for about 50 years that the presence of ADP inhibits the ATP hydrolysis reaction in the catalytic F_1_ domain of the F_o_F_1_-ATP synthase from *E. coli* [39]. Initially, ADP was considered to act as a competitive inhibitor with an inhibition constant K_i_ in the range of 300 μM. As a competitive inhibitor, ADP binds to the same catalytic binding site as ATP. Measuring the binding constants of ATP and ADP in the presence of Mg^2+^ has revealed differences and similarities between the three catalytic nucleotide binding sites on the F_1_ domain. The high-affinity MgATP binding site exhibits K_D1_ in the <10 nM range, a second medium affinity site K_D2_ around 1 μM, and a third low-affinity site K_D3_ ~20 μM [40]. For MgADP, the first high-affinity site shows a K_D1_ in the 60 nM range; for the second and third binding sites, K_D2,3_ are around 25 μM. For a nucleotide concentration of 100 μM, as used here, the nucleotide site occupancies have been determined to be almost three for MgATP (i.e., all three sites filled) and 2.5 for MgADP. These very similar binding properties support a simple competitive inhibitor model for MgADP inhibition. According to this model, an increasing MgADP fraction in a series of ADP/ATP mixtures should not affect the maximum ATP hydrolysis rate, v_max_, at high ATP concentrations, but should shift the Michaelis-Menten constant K_M_ for the half-maximal ATP hydrolysis rate to much higher ATP concentrations.

MgADP also inhibits ATP hydrolysis in the holoenzyme F_o_F_1_-ATP synthase in inverted native membrane vesicles [41,42]. At an ADP/ATP ratio of 250 μM/750 μM, the ATP hydrolysis rate drops to ~40%, and at 600 μM/400 μM, the remaining ATP hydrolysis rate is only 10%. Similarly, for purified F_o_F_1_-ATP synthase reconstituted into liposomes, the ATP hydrolysis rate drops to 44% at an ADP/ATP ratio of 250 μM/750 μM [43], or to 23% for an ADP/ATP ratio of 100 μM/100 μM [44]. Accordingly, one could estimate that an ADP/ATP ratio of 30/70 should yield a 50% inhibition of the ATP hydrolysis rate.

These biochemical assays using a few ADP/ATP ratios could not discriminate a competitive inhibitor model for MgADP from an alternative non-competitive inhibitor model. In the latter model, the inhibitor binds at a different site on the enzyme, ideally blocking turnover by trapping the enzyme-substrate complex. Consequently, the non-competitive inhibitor does not change K_M_, but lowers the maximum ATP hydrolysis rate, v_max_.

Can single-molecule FRET experiments with reconstituted F_o_F_1_-ATP synthase in the ABEL trap address and distinguish between the two models for MgADP acting as an inhibitor?

First, in contrast to regular smFRET recordings in solution, the ABEL trap increased our observation times for individual proteoliposomes to 500 ms and longer. Active, ATP-hydrolyzing single enzymes were identified by FRET state fluctuations and were clearly discriminated from inactive FRET-labeled enzymes that were characterized by a constant FRET state. An ATP turnover as slow as a few ATP molecules per second was revealed due to the long observation times. Both active and inactive enzymes were counted one after another and, therefore, changes in the fraction of active enzymes with respect to the ADP/ATP ratio were determined. We found that the mean ATP hydrolysis rates were not affected by the inhibitor ADP for a wide range of ADP/ATP ratios, but the number of active enzymes decreased with higher ADP content (Figure 7). For the case of an ideal competitive inhibitor, substrate and inhibitor will quickly bind and unbind from the same binding site. With increasing probability of binding the inhibitor, each single enzyme will slow down its turnover. However, this behavior was clearly not observed for ADP as seen in Figure 4 and Appendix A.

Second, the actual extended observation times for proteoliposomes in the ABEL trap were not long enough to discriminate the average durations of active turnover of single F_o_F_1_-ATP synthases (Figure 6 and Appendix A) and the average durations of the inactive non-rotating phases of the enzymes (Appendix A). With ATP and in the presence or absence of ADP, all conditions yielded similar time periods for the respective catalytic status of the enzymes, i.e., with 1/e times as the average between 220 and 330 ms. Previously, single-molecule rotation recordings by videomicroscopy of the isolated F_1_ domain attached to a cover glass surface provided evidence that the *E. coli* enzyme catalyzes ATP hydrolysis for about 1 s and then lapses into a transient inactive state. The inactive status lasts for 1 s [45,46], though some enzymes stay in another inactive state for about 15 s [47]. Afterwards, catalytic activity is resumed. Because the inactive state is associated with the rotary angle of the γ- and ε-subunit for the catalytic dwells, the cause of the pausing is attributed to a failed product release, or ADP inhibition, respectively [47].

While we have observed switching of the active into the inactive state of catalysis in very few enzymes, our current observation times of single F_o_F_1_-ATP synthase did not allow us to measure the time constants for the transient lapse into and out of the ADP-inhibited state. Future technical improvements of the ABEL trap recordings are possible. Changing the microfluidic chambers from PDMS/glass with a high fluorescent background to all-quartz cells will significantly increase the signal-to-noise ratio [48] and, accordingly, the internal position estimation of the trapped nanoparticle will become much more accurate. As a consequence, the feed-back voltages to push the nanoparticle into the center of the ABEL trap laser pattern will become more appropriate and the proteoliposomes are expected to be trapped tighter and much longer [28]. Furthermore, oxygen scavenger systems and a triplet quencher can be added to the buffer in order to stabilize and to prolong the photon count rates of both FRET donor and acceptor fluorophores for noise reduction of the FRET efficiency traces. Thereby the identification of fast FRET state fluctuations will become more robust. With the possibility of reaching tens of seconds of ABEL trapping time for single F_o_F_1_-ATP synthases, questions regarding the rates for transient ADP inhibition can be quantitatively addressed [25].

Third, we re-analyzed our previous single-molecule rotation data of the same FRET-labeled F_o_F_1_-ATP synthase in the ABEL trap in the presence of MgATP only [32]. This time we applied an HMM-related software approach [37] to identify FRET states and to count active as well as inactive enzymes. For the five different ATP concentrations, the percentage of non-rotating inhibited enzymes was almost constant and in the range of 70%. Despite the large variations of these numbers (Figure 7D), we did not observe an ATP concentration dependence of the inhibited fraction. Accordingly, we concluded that the increasing amounts of inhibited enzymes with increasing ADP content in the ADP/ATP mixtures indicated an increasing fraction of ADP-inhibited enzymes.

A high fraction of inhibited F_o_F_1_-ATP synthases is well known and is attributed to a mechanical blockade of subunit rotation by the C-terminus domain of the ε-subunit. The addition of LDAO (dodecyldimethylamine oxide) to the ATP hydrolysis assays increases the ATP turnover between 3- and 10-fold because the surfactant affects or dissociates the ε-subunit [49]. Using truncations of the ε-CTD allows the assessment of ADP inhibition separately from the ε-inhibition mechanism [50]. For the separated F_1_ domain of the *E. coli* enzyme, almost 90% of the F_1_ domain is ε-inhibited in the presence of the ε-subunit (with a small fraction in the ADP-inhibited state). After quantitative removal of the ε-subunit, about 90% of the F_1_ fragments are found in the ADP-inhibited state [51], and truncating the ε-CTD results in a significant increase in ATP hydrolysis rates [52,53]. Milgrom & Duncan have specified that 80% of F_o_F_1_-ATP synthases in *E. coli* membranes are inhibited, with 50% of the enzymes in the ε-inhibited form, in addition to 30% in the ADP-inhibited form [51].

It has been hypothesized that ADP inhibition and ε-CTD inhibition block the F_o_F_1_-ATP synthase at two distinct rotary angles of the γ- and ε-subunits, based on the ε-CTD-inhibited structures of F_1_ [54] and of F_o_F_1_ [34]. To examine this hypothesis by single-molecule FRET measurements with F_o_F_1_-ATP synthase, the positions of the two markers on the enzyme must enable the discrimination of all three rotary orientations for the three catalytic dwells as well as the three additional rotary positions of the ATP-waiting dwells. Our current labeling positions on residue 56 of the ε-subunit and the C-terminus of the a-subunit do not allow the discrimination of the three catalytic dwells by FRET states. Instead, our previously-used FRET acceptor position at the peripheral b-subunits would be much more suitable. If the rotary positions of catalytic dwells and ATP-waiting dwells are different, smFRET in the ABEL trap may reveal the hypothesis that ADP inhibition occurs at the rotary orientations assigned to the catalytic dwells.

Finally, ATP hydrolysis recorded with single F_o_F_1_-ATP synthase showed again a large variation in the individual turnover numbers from enzyme to enzyme called “*static disorder*” (Figure 4 and Appendix A; see [32]). To narrow these distributions, the function of the liposomes acting as a reservoir for translocated protons must be disabled. Protons are pumped into the vesicle during ATP hydrolysis. At a high H^+^ concentration, this can cause a reversal of the catalytic reaction towards ATP synthesis. Future smFRET measurements will employ lipid nanodiscs for single F_o_F_1_-ATP synthases to avoid the buildup of any proton motive force. Combination of ε-CTD truncations and using non-bleaching rotation markers like NV^−^ centers in nanodiamonds might unravel the precise conformational details of active F_o_F_1_-ATP synthases as observed one after another in the ABEL trap.

## 4. Materials and Methods

### 4.1. FRET-Labeled F_o_F_1_-ATP Synthase from E. coli

For the single-molecule FRET experiments we used a previously prepared FRET-labeled F_o_F_1_-ATP synthase reconstituted as a single enzyme in a liposome [32]. Briefly, a mutant of the F_1_ domain (*E. coli*) with a cysteine residue at position 56 of the ε-subunit was labeled with Cy3B maleimide (labeling efficiency 74%) according to published protocols [20]. The FRET donor-labeled F_1_ domain was combined with a mutant in the F_o_ domain (*E. coli*). In the F_o_ mutant, a cysteine had been added to the extended C-terminus of the a-subunit [33] and labeled with Alexa Fluor 647 maleimide as the FRET acceptor (labeling efficiency 32%). Reconstitution into pre-formed liposomes comprising phosphatidylcholine with a 5% fraction of anionic phosphatidic acid lipids was achieved according to published procedures [18,55]. Shock-frozen aliquots (5 μL) of the proteoliposomes were stored until use at −80 °C in the presence of 10% glycerol in the buffer (20 mM Tricine-NaOH (pH 8.0), 20 mM succinic acid, 50 mM NaCl, 0.6 mM KCl, 2.5 mM MgCl_2_). The acceptor-labeled a-CTD cysteine mutant of F_o_F_1_-ATP synthase solubilized in DDM yielded an ATP hydrolysis rate of 16 ATP/s at 21 °C, which increased more than tenfold by addition of LDAO [32]. The reconstituted FRET-labeled F_o_F_1_-ATP synthase achieved an ATP synthesis rate of 16 ATP/s at 21 °C [32] in good agreement with comparable FRET-labeled enzymes [20].

We measured smFRET in an ABEL trap that is based on electrophoretic and electroosmotic forces. This required a low ionic strength for the best trapping performance. We used nucleotide concentrations of 100 μM for two reasons: (1) to limit the influence of the negatively charged ATP and ADP molecules on the trapping behavior, and (2) to maintain high nucleotide concentrations almost at the level of ATP saturation for ATP hydrolysis. The KCl concentration in the buffer was set as low as possible (0.3 mM) to maintain the electrophoretic forces for controlling the position of the negatively charged proteoliposomes.

### 4.2. Recording of FRET-Labeled F_o_F_1_-ATP Synthase in a Confocal ABEL Trap

The custom-designed confocal ABEL trap microscope was described in detail before [32,56,57]. Briefly, a linearly polarized continuous-wave laser at 532 nm (Compass 315 M, Coherent, Santa Clara, CA, USA) was directed through a pair of fast electro-optical beam deflectors and attenuated to 40 μW. The laser beam was steered to a novel 60x oil immersion objective (UPlanApo with numerical aperture NA 1.50, Olympus, Tokyo, Japan) in an inverted microscope (IX71, Olympus). FRET donor fluorescence was detected in the spectral range between 545 nm and 620 nm and FRET acceptor fluorescence was detected with wavelengths >647 nm using two single-photon counting avalanche photodiodes (APD type SPCM-AQRH-14, Excelitas, Waltham, MA, USA). Photon signals were multiplexed and recorded by an external time-correlated single photon counter (TCSPC card SPC-180NX mounted in a Razer Core X PCIe box with thunderbolt 3 connection to a Win10 PC, plus 8-channel APD router HRT-82, Becker&Hickl, Berlin, Germany) and in parallel used by a field programmable gate array (FPGA card PCIe-7852R, National Instruments; in a separate computer) to calculate the transiently applied Pt-electrode feedback voltages.

The Labview-based ABEL trap software was adapted from [27] with minor modifications. We applied a 32-point knight tour laser focus pattern (focus spacing 0.47 μM) with 7 kHz repetition rate within a 2.34 × 2.34 μm area in the focal plane. Proteoliposome trapping parameters were set in the ABEL trap software using 0.6 μm focus waist, EOD deflection 0.117 μM/V, diffusion coefficient D = 5 μm^2^/s and electromobility μ = −80 µm∙s^−1^∙V^−1^. Proteoliposomes were diluted 1:600 into trapping buffer (10 mM Tricine-NaOH (pH 8.0), 10 mM succinic acid, 0.3 mM KCl, 1.25 mM MgCl_2_) with the appropriate ATP and ADP concentrations. ABEL trap sample chambers were prepared from the structured PDMS bonded to the cover glass by plasma treatment. The total volume within the ABEL trap chips was about 10 μL (for details see [32]).

### 4.3. Data Analysis of FRET-Labeled F_o_F_1_-ATP Synthase Held by the ABEL Trap

Photon counts of the FRET donor and the FRET acceptor channel were binned into 1 ms intervals using the software “Burst Analyzer” (Becker&Hickl, modified) to obtain smFRET time traces. Adding both count rates resulted in a third time trace of the so-called sum intensity. Trapping of a proteoliposome was characterized by a sudden rise of the sum signal to 30–35 counts per millisecond above the background within one time bin. Loss of the proteoliposome out of the trap was identified by a stepwise intensity drop back of the photon burst to the background. To account for the background fluctuations over time, the mean background in the time 30 ms before and after each trapping event was subtracted from each channel for each photon burst. Individual photon bursts were exported as ASCII files for further analysis in Matlab (MathWorks, Natick, MA, USA). DISC, a Matlab-based analysis package, was used to estimate and calculate the FRET states present in the proximity factor, P, time traces [37]. Determination of the FRET states was achieved in a two-step process. The first step was run in a fully unsupervised mode. In this step, the cluster division was evaluated by the Akaike Information Criterion, AIC. P time traces with more than three distinct FRET levels were re-evaluated with the Bayesian Information Criterion to avoid ambiguities.

The subsequent, second step consisted of a manual control of the idealized smFRET traces. The analysis of the idealized smFRET time traces was executed using a custom-made Matlab script. The idealized P signal was filtered using a Hempel filter over a 5-point time window. A full rotation was defined as a pair of high (P ≈ 0.8) and medium (P ≈ 0.4) FRET states. As in our previous smFRET analysis, the length of the first and of the last FRET state within a photon burst of a single F_o_F_1_-ATP synthase cannot be used because the real durations before entering and after leaving the detection volume remain unknown [18]. Therefore, calculation of FRET-based subunit rotations was done within a region of interest (ROI) defined as the time between the first and last medium-high FRET transition. The mean rotation rate per enzyme was calculated as the quotient of the number of rotations and the length of the ROI. The turnover times decay histograms were built with the lengths of each individual full rotation. The rotation time was calculated by fitting the turnover times decay histogram to a single exponential decay function with starting points at 10 ms for the ATP concentrations of 80 to 100 μM in the ADP/ATP mixtures, 14 ms for 60 and 70 μM ATP, and 16 ms for 30 to 50 μM ATP.

## Figures and Tables

**Figure 1 ijms-24-08442-f001:**
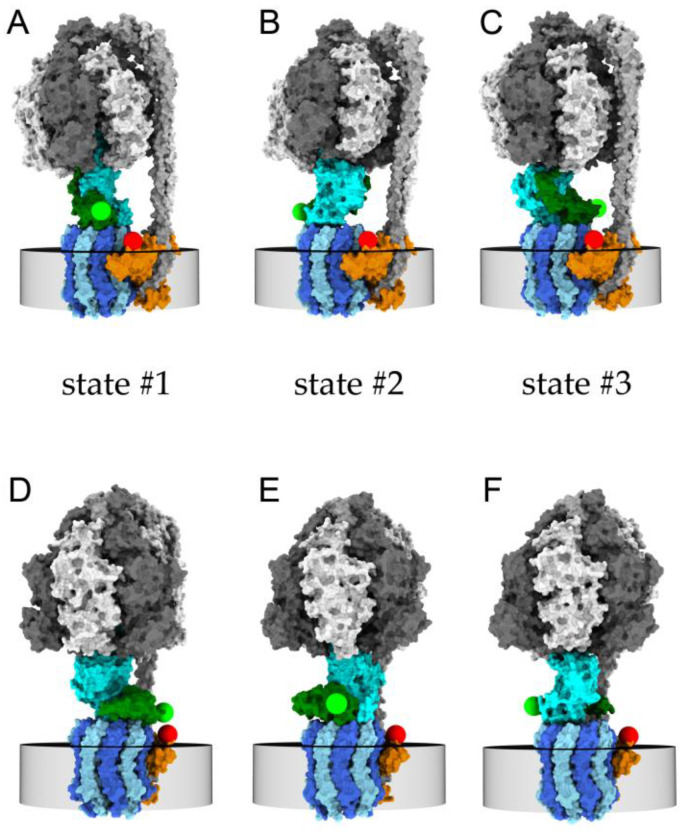
Models of FRET-labeled F_o_F_1_-ATP synthase in the three rotary states of the γ- and ε-subunits with respect to the F_1_ subunits α_3_β_3_δ (white and dark gray) and F_o_ subunits b_2_ (light gray, peripheral stalk). (**A**–**C**) cryoEM structures in the presence of 10 mM ADP with subunits b_2_ oriented to the right side [35]. (**D**–**F**) cryoEM structures in the presence of 10 mM ATP with subunits b_2_ oriented to the back side [36]. Green spheres are positions of the S atom of cysteine εH56C on the ε-subunit (dark green) labeled with FRET donor Cy3B, red spheres are modeled positions of the S atom of cysteine a276C on the a-subunit (orange) labeled with FRET acceptor Alexa Fluor 647. The central rotary γ-subunit of the F_1_ domain is shown in cyan, and c-subunits of the F_o_ domain in dark blue and in light blue, embedded in the membrane (indicated by the gray disc). During ATP hydrolysis, the rotary states move sequentially in the order …→state #1 (**D**)→state #2 (**E**)→state #3 (**F**)→… for the counter-clockwise rotation direction when viewed from the bottom.

**Figure 2 ijms-24-08442-f002:**
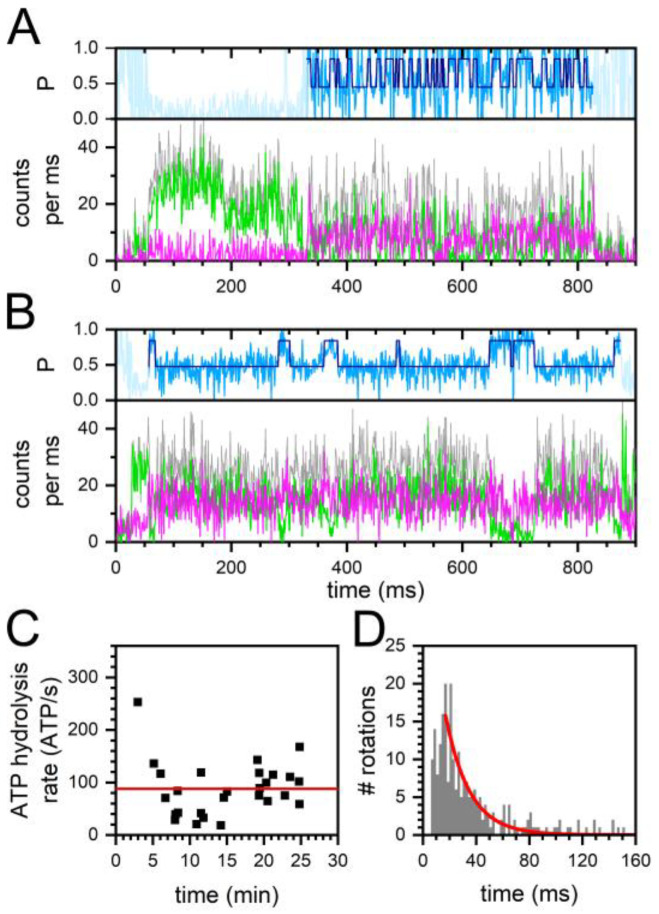
Turnover of single FRET-labeled F_o_F_1_-ATP synthases in the presence of 30 μM ADP and 70 μM ATP. (**A**,**B**) time traces of F_o_F_1_-ATP synthases with (**A**) fast and (**B**) slow ε-subunit rotation. FRET donor Cy3B photon counts per ms (green traces), FRET acceptor Alexa Fluor 647 photon counts (magenta traces), associated proximity factor P time trace in light blue, HMM-assigned FRET states in dark blue (two states). The sum intensities of FRET donor and acceptor photons are shown as gray traces. (**C**) Active enzymes with mean individual ATP hydrolysis rates over the complete 30 min recording time after addition of ADP/ATP. Mean ATP hydrolysis rate is shown as red line. (**D**) Duration of full rotations (i.e., a FRET state pair comprising one high FRET plus one subsequent low FRET state) of all active enzymes, and monoexponential decay fit (red line).

**Figure 3 ijms-24-08442-f003:**
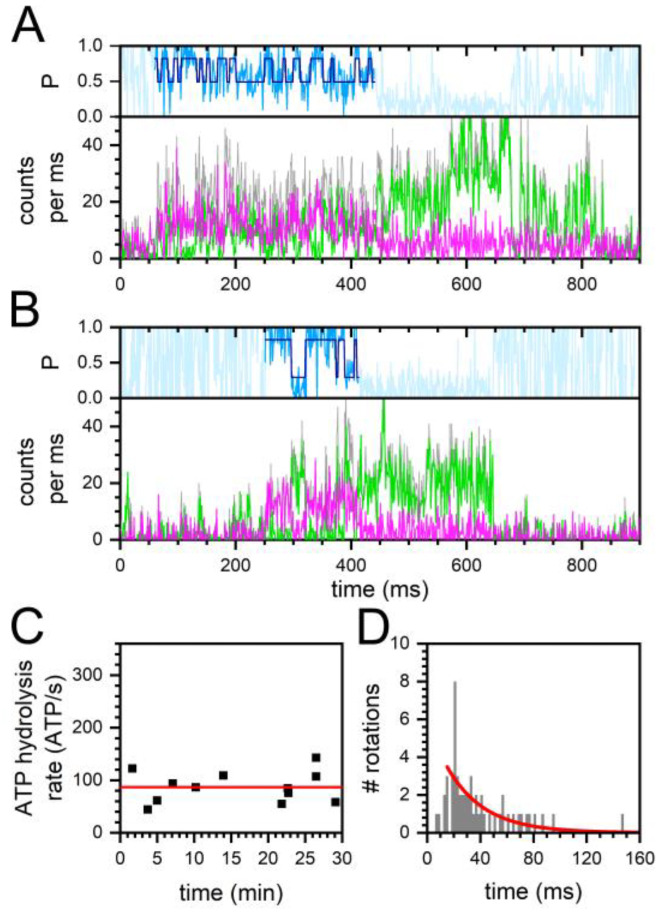
Turnover of single FRET-labeled F_o_F_1_-ATP synthases in the presence of 70 μM ADP and 30 μM ATP. (**A**,**B**) time traces of F_o_F_1_-ATP synthases with (**A**) fast and (**B**) slow ε-subunit rotation. FRET donor Cy3B photon counts per ms (green traces), FRET acceptor Alexa Fluor 647 photon counts (magenta traces), associated proximity factor P time trace in light blue, HMM-assigned FRET states in dark blue (two states). The sum intensities of FRET donor and acceptor photons are shown as gray traces. (**C**) Active enzymes with mean individual ATP hydrolysis rates over the complete 30-min recording time after addition of ADP/ATP. Mean ATP hydrolysis rate as red line. (**D**) Duration of full rotations (i.e., a FRET state pair comprising one high FRET plus one subsequent low FRET state) of all active enzymes, and monoexponential decay fit (red line).

**Figure 4 ijms-24-08442-f004:**
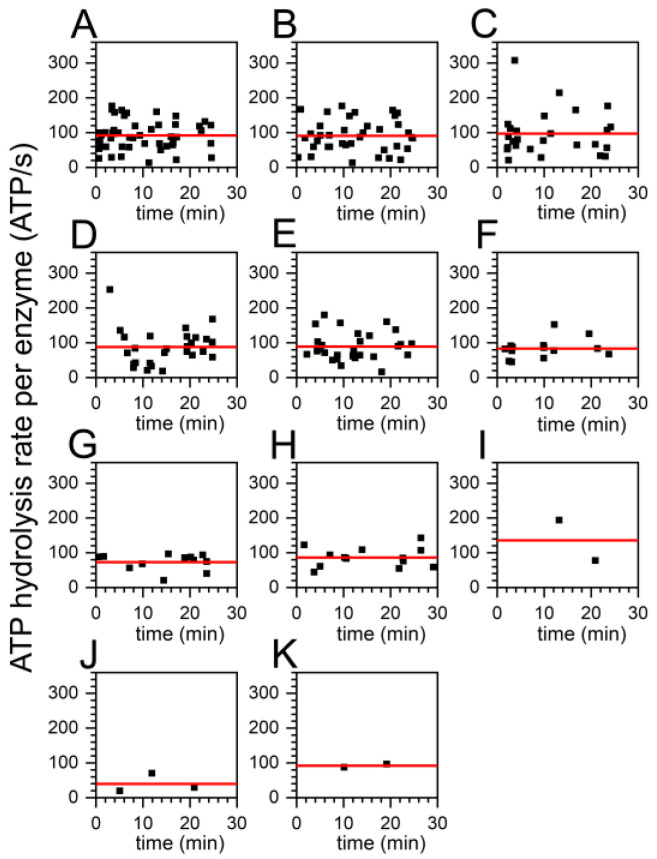
Mean turnover of individual FRET-labeled F_o_F_1_-ATP synthases (black squares) in the presence of 11 different ADP/ATP ratios as observed in the first 30 min recording time after addition of the ADP/ATP mixture. The ADP/ATP ratios were (**A**) 0/100, (**B**) 10/90, (**C**) 20/80, (**D**) 30/70 as in Figure 2C above, (**E**) 40/60, (**F**) 50/50, (**G**) 60/40, (**H**) 70/30 as in Figure 3C above, (**I**) 80/20, (**J**) 90/10, and (**K**) 100/0, nucleotide concentrations in μM. The average ATP hydrolysis rates were calculated from the mean turnover per enzyme and are shown as red lines.

**Figure 5 ijms-24-08442-f005:**
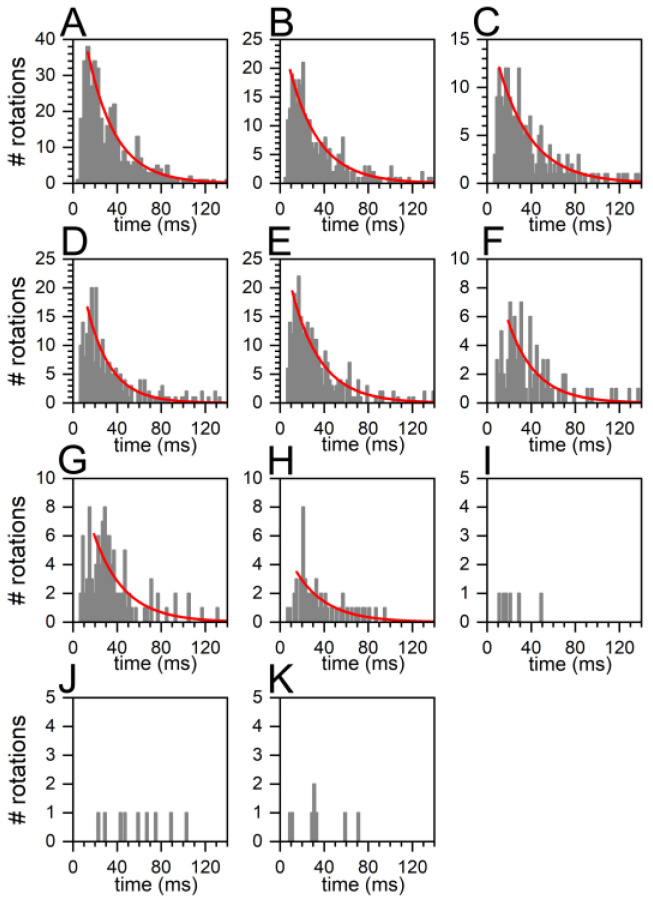
Durations of FRET state pairs in smFRET time traces comprising one high FRET plus one subsequent low FRET state. Each FRET state pair represented a full rotation of the ε-subunit of the F_o_F_1_-ATP synthases. The 11 different ADP/ATP ratios were (**A**) 0/100, (**B**) 10/90, (**C**) 20/80, (**D**) 30/70 as in Figure 2D above, (**E**) 40/60, (**F**) 50/50, (**G**) 60/40, (**H**) 70/30 as in Figure 3D above, (**I**) 80/20, (**J**) 90/10, and (**K**) 100/0, with nucleotide concentrations in μM. Histograms of FRET state pairs of all active enzymes were fitted with a monoexponential decay function (red line).

**Figure 6 ijms-24-08442-f006:**
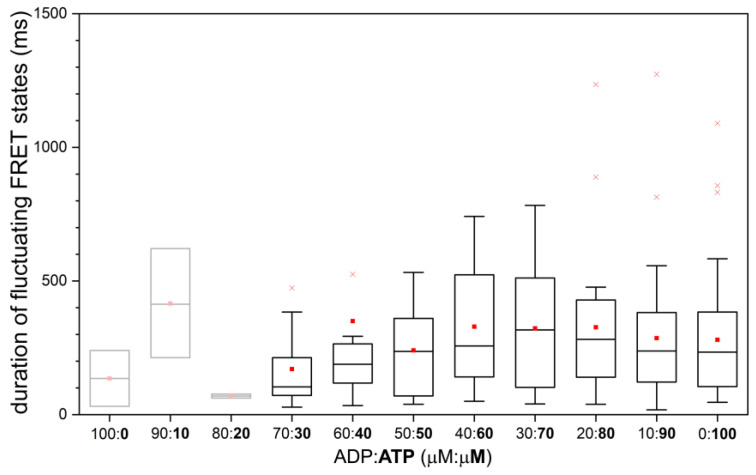
Durations of the individual fluctuating FRET state periods within the photon bursts of FRET-labeled F_o_F_1_-ATP synthases in the presence of 11 different ADP/ATP ratios. Data for the 100/0, the 90/10, and the 80/20 ADP/ATP ratios were based on less than 5 enzymes and were shaded. Red crosses were outliers, red squares represented the arithmetic mean of the fluctuating FRET state periods.

**Figure 7 ijms-24-08442-f007:**
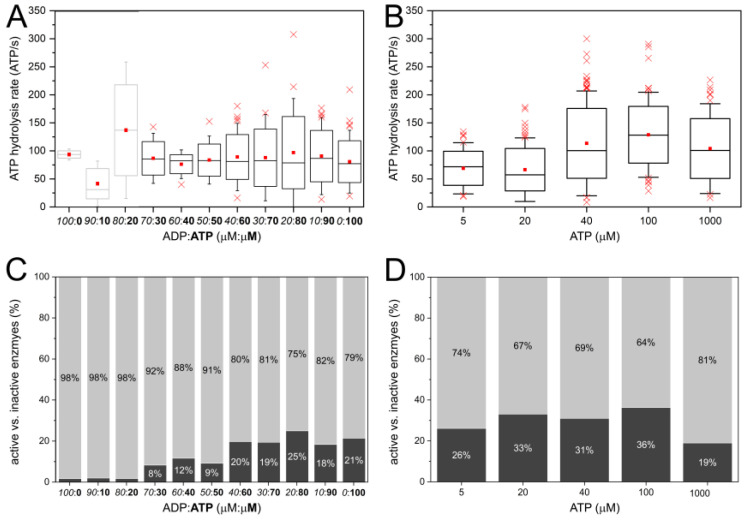
(**A**,**B**) Box plots of mean individual turnover of individual FRET-labeled F_o_F_1_-ATP synthases, (**A**) in the presence of 11 different ADP/ATP ratios and (**B**) in the presence of ATP only at five different concentrations (in μm). Red crosses were outliers, red squares were the arithmetic means. (**C**,**D**) Percentage of active (dark gray fraction) and inactive enzymes (light gray fraction), (**C**) in the presence of different ADP/ATP ratios, and (**D**) in the presence of ATP only. The data for the ATP-only figures (**B**,**D**) were obtained by re-analyzing our previous smFRET measurements [32], and by applying FRET state assignments with the DISC HMM software described in Section 4.

## Data Availability

The data presented in this study are available on request from the corresponding author.

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
