# Peer review of "Mechanism of ADP-Inhibited ATP Hydrolysis in Single Proton-Pumping F_o_F_1_-ATP Synthase Trapped in Solution"

_ijms, 2023, doi:10.3390/ijms24098442_

Round 1
Reviewer 1 Report
The manuscript "Mechanism of ADP-inhibited ATP hydrolysis in single proton-pumping FoF1-ATP synthase trapped in solution" by Pérez et al. reports new findings on the mechanism of ATPase activity regulation in ATP synthase from Escherichia coli. The authors examined single E. coli FoF1-ATP synthases in liposomes using single-molecule Forster resonance energy transfer (smFRET). They observed individual enzyme turnover by monitoring subunit rotation and used a confocal Anti-Brownian electrokinetic trap (ABEL trap) to increase observation times up to seconds.
By counting active versus inhibited enzymes, they found that increasing ADP in the ADP/ATP mixture reduced the number of remaining active enzymes operating at similar rates for varying substrate/product ratios. In summary, this study showed that ADP inhibition does not affect all FoF1-ATP synthases equally and that some enzymes are completely blocked while others continue to operate at a similar rate.
The results are new and interesting; the authors took advantage of the benefits of single biomolecule research methods and directly demonstrated the two different fractions of FoF1-ATP synthase (active and inactive). I think this study deserves publication and will be of interest to a broad circle of researchers involved in biochemical, bioenergetic and microbiological experimental work.
However, before publication, several issues have to be addressed, namely:
1) I think it is very important to verify the ATPase activity measured by smFRET by measurements with conventional biochemical methods (e.g. by measuring the Pi release). Without this control experiment it is difficult to assess the quality of the enzyme preparation and to compare the samples used in the study with the results of other research groups. It is also important to compare the turnover rates measured by smFRET with the bulk activity values.
2) It is unclear, what is the reason for inactivity of 64-98% of FoF1 molecules (Fig. 7). It is tempting to suggest that the main reason for that is ADP-inhibition or epsilon-inhibition, but the same result can theoretically be expected if during purification, labelling and reconstitution procedures 64% of the FoF molecules were irreversibly damaged. To clarify this point, I suggest to make a few measurements with LDAO or other activators that counteract the ADP-inhibition and epsilon-inhibition. It would also be very interesting to see if these activators will be more efficient at high ADP/ATP ratio.
Minor points:
i) Why was the KCl concentration so low (0.3 mM instead of physiological 150 mM)? Why only 100uM adenine nucleotides used, when the concentration in the cell is in the millimolar range? If the method used does not allow to measure under more physiological conditions, please state it in the methods section.
ii) FoF1 in liposomes generates pmf during ATP hydrolysis, and the magnitude of the pmf depends on the rate of proton leak through the membrane. Leaky liposomes will have less pmf back-pressure on the enzyme, and it will be running at a higher rate. It would be interesting to see, how addition of protonophores (e.g. FCCP or valinomycin+nigericin mixture in the presence of K+) will affect the observed turnover rates and if it will diminish the variability of the turnover rates. The phenomenon of "static disorder" might be explained by different proton permeabilities of individual liposomes.
iii) When measuring the FRET states, the authors expect to see 2 states, of which one corresponds to 2 rotational states of the enzyme. If the total duration of the FRET states is measured, is one FRET state actually ~2fold longer than the other? (One could expect this if during ATP hydrolysis FoF1 spends equal time in each rotational state)
iv) ADP-inhibition in E.coli enzyme and its peculiarities in comparison to other FoF1s were discussed in detail in "ADP-inhibition of H+-FoF1-ATP synthase. Lapashina&Feniouk, Biochemistry (Moscow) 83 (10), 1141-1160". The authors might find this paper useful for the Discussion section.
v) In lines 62-63 the authors mention the role of subunit gamma as the "director" of rotary catalysis. However, it seems that the alpha3beta3 hexamer itself is doing the rotary catalysis well enough (see https://doi.org/10.1126/science.1205510 )
After these issues are addressed, I think the paper can be published. I also have to point out that the method used provides the authors with a great tool to further investigate the regulatory mechanisms of ATP synthase, so I hope to see more interesting experimental works from this group in the future.
Reviewer 2 Report
Reviewer’s Comments:
The manuscript “Mechanism of ADP-inhibited ATP hydrolysis in single proton-pumping FoF1-ATP synthase trapped in solution” is a very interesting work. In this work, FoF1-ATP synthases in mitochondria, in chloroplasts and in most bacteria are the proton-driven membrane enzymes supplying the cells with ATP made from ADP and phosphate. To monitor and prevent the reverse chemical reaction of fast wasteful ATP hydrolysis by the enzymes, different control mechanisms exist including mechanical or redox-based blockade of catalysis and ADP inhibition. In general product inhibition is expected to slow down the mean catalytic turnover. However, biochemical assays are ensemble measurements and cannot discriminate between a mechanism affecting all enzymes equally or individually. For example, all enzymes could work slower at a decreasing substrate/product ratio, or more and more individual enzymes are blocked completely. Here, we examined how increasing amounts of ADP affected ATP hydrolysis of single Escherichia coli FoF1-ATP synthases in liposomes. We observed individual catalytic turnover of the enzymes one after another by monitoring the internal subunit rotation using single-molecule Förster resonance energy transfer (smFRET). Observation times of single FRET-labeled FoF1-ATP synthase in solution were increased up to seconds using a confocal Anti-Brownian electrokinetic trap (ABEL trap). While I believe this topic is of great interest to our readers, I think it needs major revision before it is ready for publication. So, I recommend this manuscript for publication with major revisions.
1. In this manuscript, the authors did not explain the importance of ATP hydrolysis in the introduction part. The authors should explain the importance of ATP hydrolysis.
2) Title: The title of the manuscript is not impressive. It should be modified or rewritten it.
3) Correct the following statement “By counting active versus inhibited enzymes we revealed that ADP inhibition did not decrease the catalytic turnover of all FoF1-ATP synthases equally. Instead, increasing ADP in the ADP/ATP mixture reduced the number of the remaining active enzymes which were operating at similar catalytic rates for varying substrate/product ratios”.
4) Keywords: The ATP hydrolysis is missing in the keywords. So, modify the keywords.
5) Introduction part is not impressive. The references cited are very old. So, Improve it with some latest literature like 10.1016/j.molstruc.2021.131136, 10.3390/molecules27217368
6) The authors should explain the following statement with recent references, “As discussed above, the ADP/ATP ratios 80/20, 90/10 and 100/0 were excluded from this analysis due to the very low number of active enzymes in these data sets”.
7) Add space between magnitude and unit. For example, in synthesis “21.96g” should be 21.96 g. Make the corrections throughout the manuscript regarding values and units.
8) The author should provide reason about this statement “the box plot is shown in Figure 6 and the scatter plot for these data is found in the supporting information as Figure S2”.
9. Comparison of the present results with other similar findings in the literature should be discussed in more detail. This is necessary in order to place this work together with other work in the field and to give more credibility to the present results.
10) Conclusion part is very long. Make it brief and improve by adding the results of your studies.
11) There are many grammatic mistakes. Improve the English grammar of the manuscript.
Minor editing of English language required
Round 2
Reviewer 1 Report
Most of my comments were adequately dealt with; I think the paper can be published.